epidemiology; genetics; global mental health; mental disorder; prevalence and risk factors

**Corresponding author:**
Kristin Schmidt;
Email: krisschmidt@udec.cl

# Latent classes of mental health disorders and their associations with polymorphisms of 5HTTLPR and BDNF in a Chilean primary care population

Esteban Moraga-Escobar[1], Benjamín Vicente[1,2], Romina Rojas-Ponce[1,3,4] ⓘ, Juan Luis Castillo-Navarrete[1,4,5], Alejandra Guzman-Castillo[1,4,6], Ximena Macaya[1,2], Paola Lagos Muñoz[1,3] and Kristin Schmidt[1,2] ⓘ

[1]Programa Neurociencias, Psiquiatría y Salud Mental, NEPSAM, Universidad de Concepción, Concepción, Chile; [2]Departamento Psiquiatría y Salud Mental, Facultad de Medicina, Universidad de Concepción, Concepción, Chile; [3]Departamento de Farmacología, Facultad de Ciencias Biológicas, Universidad de Concepción, Concepción, Chile; [4]Programa Doctorado en Salud Mental, Departamento Psiquiatría y Salud Mental, Universidad de Concepción, Concepción, Chile; [5]Departamento Tecnología Médica, Facultad de Medicina, Universidad de Concepción, Concepción, Chile and [6]Departamento Ciencias Básicas y Morfología, Facultad de Medicina, Universidad Católica de la Santísima Concepción, Concepción, Chile

## Abstract

This study explored the association between serotonin transporter gene (5HTTLPR) and brain-derived neurotrophic factor gene (BDNF) polymorphisms with mental health disorders in a Chilean primary care population using latent class analysis. The sample included 789 adults genotyped for 5HTTLPR and BDNF, who were assessed for psychiatric diagnoses using the Composite International Diagnostic Interview (CIDI). Two distinct mental health profiles emerged: a high psychiatric comorbidity group, marked by a high prevalence of anxiety and stress-related disorders, and a low comorbidity group. The study found that the L'/L' polymorphism of the serotonin transporter gene was associated with a reduced risk of belonging to the high-comorbidity group, particularly when paired with the GG polymorphism of the BDNF gene. These findings suggest a synergistic interaction between these genes that influences susceptibility to psychiatric disorders. This research underscores the importance of considering genetic interactions in mental health studies and highlights the utility of latent class analysis in identifying clinically relevant diagnostic profiles, which could enhance early detection and intervention strategies in primary care.

## Impact statements

This study is one of the few national-level investigations that utilised a large sample size from the Chilean primary care population to explore the relationship between genetics and mental health profiles through latent class analysis. By applying this innovative methodology, this study provides a model for future studies aimed at unravelling the complex nature of psychiatric conditions. The findings have implications beyond Chile, challenging existing mental health diagnostic systems and highlighting the importance of evaluating the interaction between different genetic polymorphisms.

This study demonstrates that the interaction between polymorphisms influences the likelihood of belonging to groups with varying levels of psychiatric comorbidity. Specifically, a reduced risk was observed with the L'/L' serotonin transporter 5HTTLPR polymorphism but only in the presence of the GG BDNF polymorphism. This interaction may help to explain inconsistent results found in previous studies that examined single genes in isolation without considering the broader genetic context.

The methodological insights from this study could significantly benefit future research in the genetic–mental health field. Additionally, these findings have important implications for improving mental health outcomes through personalised medicine. By identifying specific genetic profiles linked to varying psychiatric comorbidities, health professionals can develop more targeted screening and intervention strategies, particularly in primary care settings where early detection and treatment are crucial.

## Introduction

Individual variations in mental health may be linked to genetic polymorphism patterns (Nestor et al., 2021b). Owing to the significant disease burden and heritability of mental disorders

(Burmeister et al., 2008; Gatt et al., 2015), numerous studies have sought to identify genetic associations to identify specific genes that may serve as risk factors for these conditions (Gatt et al., 2015; Mei et al., 2021). One stream of research has concentrated on allelic variants of single nucleotide polymorphisms, positing that these variations affect neurotransmitter neurotrophic functions (Nestor et al., 2021b), thereby influencing the occurrence of mental disorders.

Among the most well-known and extensively studied polymorphisms are those in the serotonin transporter gene (5HTTLPR), which regulates serotonin transport from the synaptic to the pre-synaptic space in neurons (Oo et al., 2016), thereby affecting its availability. The biallelic polymorphism of the SLC6A4 gene on chromosome 17q11.2, due to a 44bp insertion/deletion in the 5' regulatory region, results in a long L allele (with the insertion) and a short S allele (with the deletion). Carriers of the S allele exhibit lower transcriptional levels of this transporter (Fratelli et al., 2020; Miozzo et al., 2020). Additionally, single nucleotide polymorphisms (rs25531 and rs25532) create different allelic subtypes regarding the functionality of the AP2 transcriptional regulator, corresponding to the S, Lg (similarly functional to S), and La alleles (Fratelli et al., 2020). Another extensively studied polymorphism is the brain-derived neurotrophic factor (BDNF) gene, which is essential for regulating adult brain plasticity (Gören, 2016). The most investigated BDNF polymorphism is the single nucleotide polymorphism, Val66Met (Nieto et al., 2021). This functional polymorphism results from an amino acid substitution from valine (Val) to methionine (Met) at codon 66 of the 5' pro region of BDNF, disrupting interactions with intracellular transport molecules and affecting the regulation of BDNF release at the synapse (Notaras et al., 2015). This disruption interferes with processes involving this neurotrophin, including neuronal plasticity, neurotransmission modulation, and the survival of dopaminergic, cholinergic and serotonergic neurons, among others (Gören, 2016).

These genetic variations have been investigated as potential risk factors for mental disorders, showing associations with conditions like schizophrenia (Autry and Monteggia, 2012; Gatt et al., 2015; Notaras et al., 2015; Gören, 2016; Di Carlo et al., 2019; Nieto et al., 2021), mood disorders (Gatt et al., 2015; Notaras et al., 2015; Gören, 2016; Oo et al., 2016; Fratelli et al., 2020; Miozzo et al., 2020), anxiety disorders (Autry and Monteggia, 2012; Notaras et al., 2015; Miozzo et al., 2020; Tanahashi et al., 2021), suicidal behaviour (Fanelli and Serretti, 2019; Fratelli et al., 2020), obsessive-compulsive disorder (OCD) (Mak et al., 2015; Miozzo et al., 2020), substance use disorders (Oo et al., 2016) and eating disorders (Autry and Monteggia, 2012; Notaras et al., 2015). However, the results remain inconclusive and inconsistent, even for the same genetic variation. For instance, Miozzo *et al.* (Miozzo et al., 2020) identified the short allele of the serotonin transporter as a protective factor for OCD, whereas (Mak et al., 2015) linked the same allele to an increased risk of OCD in females. Likewise, (Miozzo et al., 2020) associated the short allele with panic disorder, whereas (Tanahashi et al., 2021) found higher anxiety levels in long allele (L') carriers compared to short allele (S'/S') carriers in panic disorder without agoraphobia. Regarding BDNF and schizophrenia, the Val66Met polymorphism has been proposed as a risk factor (Gatt et al., 2015; Notaras et al., 2015; Gören, 2016). Nonetheless, findings remain inconclusive, with some studies suggesting that it is a protective factor (Di Carlo et al., 2019), others showing no association with diagnosis but influencing the age of onset (Gören, 2016), or indicating that its effect depends on the homozygous status of the risk polymorphism (Val/Val or Met/Met) (Gatt et al., 2015).

The absence of definitive outcomes may be attributed to the inadequacy of straightforward association analyses between a specific genetic alteration and a particular psychiatric disorder, particularly given the intricate nature of genetic and psychiatric research (Burmeister et al., 2008). To address this situation, authors have considered the interaction of both polymorphisms with the presence of mental health pathologies, given their established association with similar disorders. Terracciano et al. (2010), analysed the association of BDNF with neuroticism, a risk factor for mental disorders, both independently and in relation to its interaction with 5HTTLPR. They discovered that, while BDNF alone was not significantly linked to neuroticism, differences emerged in the group of LL allele carriers of 5HTTLPR. Specifically, Val variant carriers of BDNF had lower neuroticism scores, whereas Met variant carriers had higher neuroticism scores. These variations, which appear when considering the interaction between the two genes, have also been observed in university students' psychopathological characteristics. A protective effect of the Met variant of BDNF, indicated by lower scores on various psychopathological dimensions, was only evident in those who did not carry the S' variant of 5HTTLPR (Kourmouli et al., 2013). Furthermore, a protective effect of L-homozygosity of 5HTTLPR on neuroticism scores was found only in carriers of the Met variant of the BDNF Val66Met polymorphism in a previous Chilean study that included patients with borderline personality disorder (Salinas et al., 2020). The interaction between different genes, known as epistasis, is of particular concern because if the effect of one genetic change is altered or masked by the effects of another, the ability to detect the association with a disease is likely to be reduced (Cordell, 2002). This genetic overlap that would imply a common risk for certain diagnoses could be understood from the perspective that genetics generate vulnerability profiles to certain stressors, leading to differences in mental health among individuals (Nestor et al., 2021a). Stressors can be varied, including those associated with traumatic events and/or influenced by environmental, psychosocial or biological characteristics (Chu et al., 2025).

These differences in mental health between individuals, when understood as differences in the presence of mental health disorders, also present analytical challenges, including the overlap of certain symptoms and a high degree of comorbidity between different disorders (Mei et al., 2021). It is crucial to acknowledge that the current diagnostic criteria are based on expertly defined standards (Figueroa, 2019), which may not necessarily reflect a definitive etiological basis for diagnosis. This is evidenced by the fact that diagnostic categories established through clinical consensus do not always align with findings from clinical neuroscience and genetics (Insel et al., 2010). Considering the symptomatic overlap and comorbidity among diagnoses and the inconclusive results from analysing individual mental health disorders, it may be advantageous to examine the genetics–mental health relationship based on diagnostic profiles encompassing multiple disorders. However, the numerous existing diagnoses complicate this approach. A potential solution involves using methods, such as latent class analysis, to handle these variables and create mental disorder profiles. Latent class analysis is a statistical technique used to identify qualitatively distinct subgroups within a heterogeneous population that shares certain common characteristics among their members (Weller et al., 2020). This person-centred approach provides insight into the interaction of numerous different variables and their combinations to create different profiles (Göbel and Cohrdes, 2021). In this way, different authors have been able to examine the profiles of their populations based on multiple mental health risk factors and

behaviours (Parra et al., 2006; Göbel et al., 2016; Göbel and Cohrdes, 2021; Hu et al., 2021), as well as profiles related to mental health disorders (Nelon et al., 2019; Liu et al., 2021). For example, Liu et al. (2021) were able to go from 16 mental health symptoms to three profiles of anxiety and depression in medical students during the COVID-19 epidemic in China: a low psychological well-being group, a moderate psychological well-being group and a low anxious-depressive symptoms group (good psychological well-being). Göbel and Cohrdes (2021) reduced a set of 27 mental health risk factors to a four-class solution (basic risk, parent risk, social risk and high risk). This allowed the authors to ascertain risk factors for more than one diagnosis (Liu et al., 2021) and delineate profiles within a heterogeneous group, thereby facilitating the formulation of prevention or intervention strategies for an identified risk profile (Göbel and Cohrdes, 2021; Liu et al., 2021).

In view of the above, the present study aimed to evaluate the association of the interaction between genetic polymorphisms of 5HTTLPR and BDNF with a profile of different mental health disorders, based on the use of latent class analysis, in a Chilean primary care population.

## Methods

### Design and sample

The data analysed were part of a prospective cohort of several waves of mental health surveys of adults attending primary care centres in the province of Concepción, Chile. The centres were part of the Chilean National Public Health Service, representative of rural and urban facilities and stratified by socioeconomic status, totalling 10 centres. Regardless of the reason for their visit (e.g., medical check-up, vaccination, information, etc.), patients were invited to participate at their respective centre. Exclusion criteria were the presence of acute psychosis, dementia, and/or a disabling physical illness, and inability to understand the local language.

The initial study (FONDEF study no. DO2I-1140) was conducted in 2003 with a random sample of 2878 adults aged 18-75 years being recruited. After informed consent was obtained, trained interviewers administered the questionnaires through face-to-face interviews. A second study (FONDECYT N°1110687) was carried out between 2011 and 2012, in which participants recruited in the first study were contacted to provide saliva samples that could be processed for genotyping. Of these, 937 individuals were correctly genotyped for both 5HTTLPR and BDNF. The studies were approved by the Research Ethics Committee of the University of Concepción, Faculty of Medicine. Further details of the study design and procedures have been reported elsewhere (Rojas et al., 2015).

### Variables

#### Mental health diagnoses

The presence or absence of psychiatric disorders was determined using the World Health Organization (WHO) Composite International Diagnostic Interview (CIDI). The CIDI is a structured interview that provides recent and lifetime psychiatric diagnoses according to the International Classification of Diseases, 10th edition (ICD-10). The CIDI has been demonstrated to be a highly valid instrument for the diagnosis of psychiatric disorders, with the official Spanish version having been previously validated in studies of a Chilean population (Vicente et al., 2002; Vicente et al., 2006).

#### Sociodemographic and psychosocial

The sociodemographic variables considered in this study were sex, age and educational level, which were divided into three categories (basic education or less, high school and higher education). Two psychosocial variables were included: the presence of a family history of depression and the number of different forms of abuse experienced in childhood, including physical, psychological and sexual abuse.

### Genetics

#### DNA extraction

Saliva samples were obtained, preserved and transported using a DNA collection kit (Oragene-DNA G-500; DNAgenotek®, Canada). DNA extraction was performed using the salt precipitation method. DNA concentration was quantified using an Infinite® 200 PRO NanoQuant spectrophotometer (Tecan, Switzerland). Finally, DNA integrity was confirmed using agarose gel electrophoresis.

#### Val66/met BDNF genotypification

Typification was performed using restriction enzyme-based PCR (BsaA I). The PCR fragments were digested with the restriction enzyme BsaA I (New England Biolab, MA, USA). This enzyme produces three fragments of 275, 321 and 77 bp when guanine is present at nucleotide 1249. Conversely, when cytosine was present at this position, two fragments of 321 and 352 bp were produced. Finally, the digested PCR products were separated on 1.2% agarose gel. The final classification of the functional genotypes derived from the allele combination were classified as G/G (valine/valine), G/A (valine/methionine) and A/A (methionine/methionine).

#### 5-HTTLPR genotypification

Genotyping of the 5-HTTLPR gene was conducted via polymerase chain reaction (PCR) to identify the presence of short and long alleles (28,31). The alleles were amplified using the following primers: sense F1 (5′-TCCTCCGCTTTGGCGCCTCTCTTCG-3′) and antisense R1 (5′-TGGGGGGTTGCAGGGGGGAGATCC TG-3′). The resulting products were 469 and 512 bp for the short and long alleles, respectively. Subsequently, the PCR fragments were digested with MspI restriction enzyme (New England Biolabs, MA, USA). This resulted in the generation of distinct cut patterns, including SA (469 bp), SG (402 bp and 67 bp), LA (512 bp) and LG (402 and 110 bp). Finally, the resulting fragments were visualised on 3% agarose gel. All genotyping reactions were conducted in duplicate. The final classification of the functional genotypes derived from the allele combination was as follows: L'/L' (which included only LA/LA combinations), L'/S' (LA/SA, LA/SG, LA/LG) and S'/S' (SA/SA, SG/SA, SG/SG, SA/LG, SG/LG, SG/LG).

#### Val66Met and 5-HTTLPR polymorphism analysis

To analyse the relationships between these functional genotypes, comparison groups were established. The allele combinations of each gene were divided into higher-risk and lower-risk groups for psychiatric disorder development based on literature references and transcriptional/secretory activity. The genotypes associated with reduced activity – A/A for Val66Met and S'/S' for 5-HTTLPR – were classified as higher risk. Conversely, the G/G genotype for Val66Met and the L'/L' genotypes for 5-HTTLPR were categorised as lower risk. Heterozygous genotypes (G/A for Val66Met and L'/S' for 5-HTTLPR) were included in the higher-risk groups in the

regression analysis due to the presence of a lower-activity allele potentially elevating psychiatric disorder risk.

### Data analysis

Data were analysed using R version 4.0.0. A review of missing cases identified a range of missing data for different mental health diagnoses, from 0.1 to 4.2%. Consequently, these cases were eliminated, leaving a final sample of 789 participants with the required variables for inclusion in this study.

The normality of the numerical variables was assessed using graphical methods and the Lilliefors test, with the results presented for descriptive purposes as mean (standard deviation) or median (interquartile range). Bivariate analyses were performed using the t-test or Kruskal–Wallis test, respectively. Categorical variables are presented as absolute and relative frequencies. Differences were evaluated using the chi-squared test. All statistical significance tests were conducted using two-sided tests at a significance level of 5%.

### Latent class analysis

Latent class analysis was conducted using the poLCA package. Furthermore, because there were no cases of delusional disorders in this study, this diagnosis was not included in the final models.

Subsequently, a series of latent class models with an increasing number of classes was examined to select the final model using statistical and theoretical criteria. Table 1 presents the results of statistical criteria based on the number of classes included in the models.

The two-class solution was selected on the basis of both optimal statistics and interpretability, particularly in view of the lower Bayesian information criterion (BIC), which is considered the most reliable indicator of model fit (Weller et al., 2020). This solution yielded two clinical profiles that are reasonable by the literature and that will be reviewed in the discussion section. In addition, it ensured sufficient sample sizes for the remaining analyses.

### Regressions

We then examined the independent associations between genetic predictors and the risk of belonging to a higher psychiatric comorbidity profile by using univariate logistic regression analysis with logit linkage. The largest sample size for each functional polymorphism was designated as the reference group. To test the association between all variables (genetic, biological and psychosocial) and the risk of having a higher psychiatric comorbidity profile, a series of hierarchical multivariate logistic regression analyses were performed. The first two models included only genetic risk factors, independently and with their interaction. The next model included additional biological variables, and the final model included psychosocial factors. This approach was adopted to adjust for the influence of these additional variables on the associations identified between the two genes.

**Table 1.** Parameters of the different LCAs

| Classes | AIC | BIC | $G^2$ | $Chi^2$ | DF |
|---|---|---|---|---|---|
| **2** | **6300** | **6473** | **819** | **168100** | **37** |
| 3 | 6278 | 6540 | 759 | 84768 | 56 |
| 4 | 6227 | 6577 | 670 | 309626 | 75 |
| 5 | 6231 | 6670 | 636 | 281339 | 94 |

The interactions of polymorphisms in these models were evaluated using a multiplicative model; however, interaction was also evaluated from an additive perspective using relative excess risk due to interaction (RERI), as has been done in other similar studies (Rozenblat et al., 2017). When an interaction is present in the data, RERI will be greater than 0.

Subsequently, a post-hoc analysis of the serotonin transporter association was performed based on the findings of polymorphism interactions. This analysis was conducted using separate samples defined by BDNF polymorphisms to facilitate a more comprehensive understanding of the observed variation in the association.

## Results

### Descriptives

Of the 2,878 participants interviewed in the first wave of the cohort, 1,223 were contacted to provide saliva samples for genotyping, of which 937 individuals were correctly genotyped for 5-HTTLPR and BDNF. After excluding those with missing data for analysis, a total of 789 participants remained, which constituted the final sample. 77.8% were female, with an average age of 47.6 years (SD = 16.5). Only one person was not Chilean (Jamaican) and 16 participants identified as belonging to an indigenous group. With regard to functional genotypes, 50.2% of the sample exhibited a GG genotype in BDNF, while only 16.3% displayed a L'/L' genotype of 5HTTLPR. The remaining sociodemographic variables and polymorphisms in the sample are shown in Table 2.

### Latent class analysis

Figure 1. A illustrates the probabilities profiles of various mental health diagnoses associated with the two latent classes. The first class, representing 20.4% of the sample, was more likely to present multiple mental health disorders, with higher probabilities for specific phobia, post-traumatic stress disorder and somatoform disorders. It had a lower probability of bipolar depression, psychotic disorders, mania and panic disorders, while the remaining diagnoses exhibited probabilities close to 20%. In contrast, the second class was more likely to have simple depression and specific phobias, each with a probability close to 20%, while other diagnoses were much less likely. The lifetime prevalence of each individual diagnosis for the total sample and each group can be found in Supplementary Appendix 1.

The most notable differences between the probabilities of both groups were concentrated in anxiety disorders (agoraphobia, generalised anxiety, social phobia and specific phobias), dissociative disorders, chronic depression (recurrent depression and dysthymia), PTSD and somatoform disorders, with a higher probability of occurrence in latent class 1. Upon analysis of the prevalence of multiple psychiatric comorbidities in latent class 1, a significant difference was identified in the median number of disorders compared to its counterpart, with 4 (interquartile range (IQR) = 3-5) for this group versus 1 (IQR = 0-1) for latent class 2 (p < 0.001) (Table 2). Based on these results and throughout the article, the group identified as latent class 1 will be referred to as the 'Higher psychiatric comorbidity' group, and the group identified as latent class 2 will be referred to as the 'Lower psychiatric comorbidity' group. The higher psychiatric comorbidity group was characterised by a higher prevalence of female participants (p < 0.001), a greater number of different forms of abuse experienced in childhood (p < 0.001) and a higher prevalence of family history of depression (p < 0.001; Table 2).

**Table 2.** Characteristics of polymorphisms, socio-demographic variables and number of disorders in the total sample and in the latent classes

| | | Total (N=789) | Latent Class 1 (Higher psychiatric comorbidity) (n=161) | Latent Class 2 (Lower psychiatric comorbidity) (n=628) | p |
|---|---|---|---|---|---|
| BDNF functional genotype | GG | 396 (50.2) | 80 (49.7) | 316 (50.3) | 0.631 |
| | AA | 351 (44.5) | 70 (43.5) | 281 (44.7) | |
| | GA | 42 (5.3) | 11 (6.8) | 31 (4.9) | |
| 5HTTLPR functional genotype | L'/L' | 129 (16.3) | 17 (10.6) | 112 (17.8) | **0.022** |
| | L'/S' | 393 (49.8) | 94 (58.4) | 299 (47.6) | |
| | S'/S' | 267 (33.8) | 50 (31.1) | 217 (34.6) | |
| BDNF+5HTTLPR Interaction | GA-AA + L'/L' | 69 (8.7) | 11 (6.8) | 58 (9.2) | 0.129 |
| | GA-AA + L'/S'-S'/S' | 324 (41.1) | 70 (43.5) | 254 (40.4) | |
| | GG + L'/L' | 60 (7.6) | 6 (3.7) | 54 (8.6) | |
| | GG + L'/S'-S'/S' | 336 (42.6) | 74 (46.0) | 262 (41.7) | |
| Sex | Female | 614 (77.8) | 149 (92.5) | 465 (74.0) | **<0.001** |
| | Male | 175 (22.2) | 12 (7.5) | 163 (26.0) | |
| Age | Mean (SD) | 47.6 (16.5) | 47.3 (14.5) | 47.7 (17.0) | 0.781 |
| Educational level | Basic education or less | 436 (55.3) | 85 (52.8) | 351 (55.9) | 0.499 |
| | High school | 287 (36.4) | 59 (36.6) | 228 (36.3) | |
| | Higher education | 66 (8.4) | 17 (10.6) | 49 (7.8) | |
| Number of different forms of abuse experienced in childhood[a] | 0 | 425 (53.9) | 56 (34.8) | 369 (58.8) | **<0.001** |
| | 1 | 157 (19.9) | 34 (21.1) | 123 (19.6) | |
| | 2 | 160 (20.3) | 50 (31.1) | 110 (17.5) | |
| | 3 | 47 (6.0) | 21 (13.0) | 26 (4.1) | |
| Presence of a family history of depression | No | 611 (77.4) | 100 (62.1) | 511 (81.4) | **<0.001** |
| | Yes | 178 (22.6) | 61 (37.9) | 117 (18.6) | |
| Number of mental health disorders | Median (IQR) | 1.0 (0.0 to 2.0) | 4.0 (3.0 to 5.0) | 1.0 (0.0 to 1.0) | **<0.001** |

[a]Physical, psychological and/or sexual abuse (scale (sum) from 0 to 3).

The prevalence of the different diagnoses in relation to the number of comorbidities identified by each latent class can be seen in part B of Figure 1. It should be noted that in the group with the highest psychiatric comorbidity, no cases were identified with only one diagnosis or less. This contrasts with the group with the lowest psychiatric comorbidity, in which 45.5% of the sample had no diagnosis at all during their lifetime (n = 286).

## Associations

Table 3 presents the logistic regression models for the polymorphisms and their interactions with membership in the group with the highest psychiatric comorbidity. In the univariate model, a statistically significant reduction in the risk of belonging to this group (OR 0.54, 95% CI 0.31-0.91, p = 0.028) was identified in individuals presenting with the 5HTTLPR functional genotype L'/L' compared to those who did not, with no association found in relation to the BDNF functional genotype in this model.

When considering the interaction between these genotypes, we can conclude that the reduction in risk of L'/L' is only statistically significant in individuals who possess the BDNF GG functional genotype. This significance and interaction remained when additional

biological variables (sex and age) and psychosocial variables (schooling, family history of depression and number of childhood forms of maltreatment) were included in the analysis. This was also tested by running post hoc regressions using subsamples based on BDNF polymorphisms (Table 4). The evaluation of the interaction on the additive scale after recoding for risk factor yielded a relative excess risk due to interaction (RERI) of GA-AA x L'/S'-S'/S' of -1.27 (95% CI: -3.95 – 0.141; p = 0.82322).

In addition, the model revealed a significant association between sex and belonging to the group with the highest psychiatric comorbidities. Male sex was found to be protective (OR 0.23; 95%CI, 0.12-0.42; p < 0.001) compared to female sex. A family history of depression was also identified as a risk factor (OR, 2.48; 95% CI, 1.65-3.72; p < 0.001). Finally, the model demonstrated a significant association between the number of different forms of childhood maltreatment and the outcome.

In order to facilitate the visualisation of the influence of the interaction of polymorphisms on the proportion of people who belonged to the group with the highest psychiatric comorbidity, Supplementary Appendix 2 has been produced. This figure shows the increase in the proportion of the group when going from L'/L' to L'/S'-S'/S' based on the BDNF polymorphism.

(A)

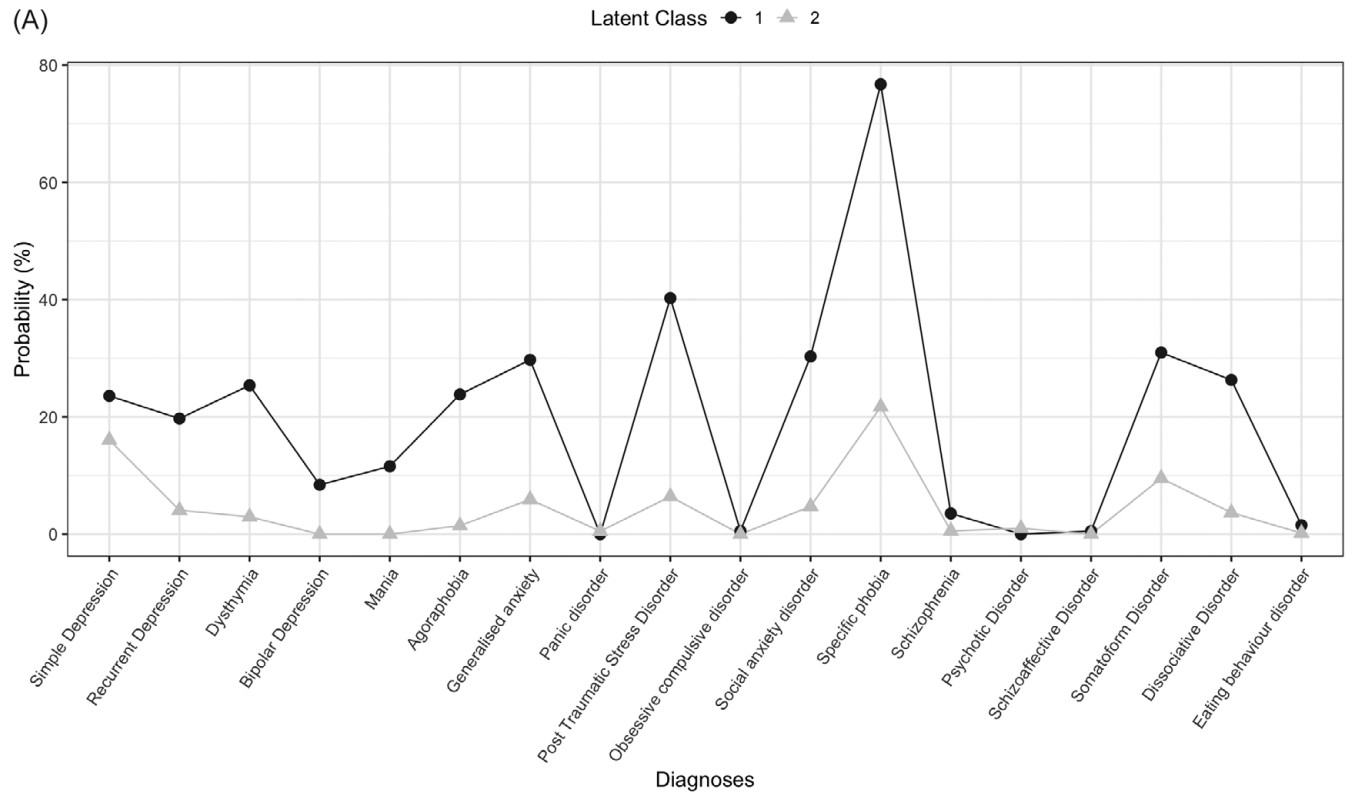

(B)

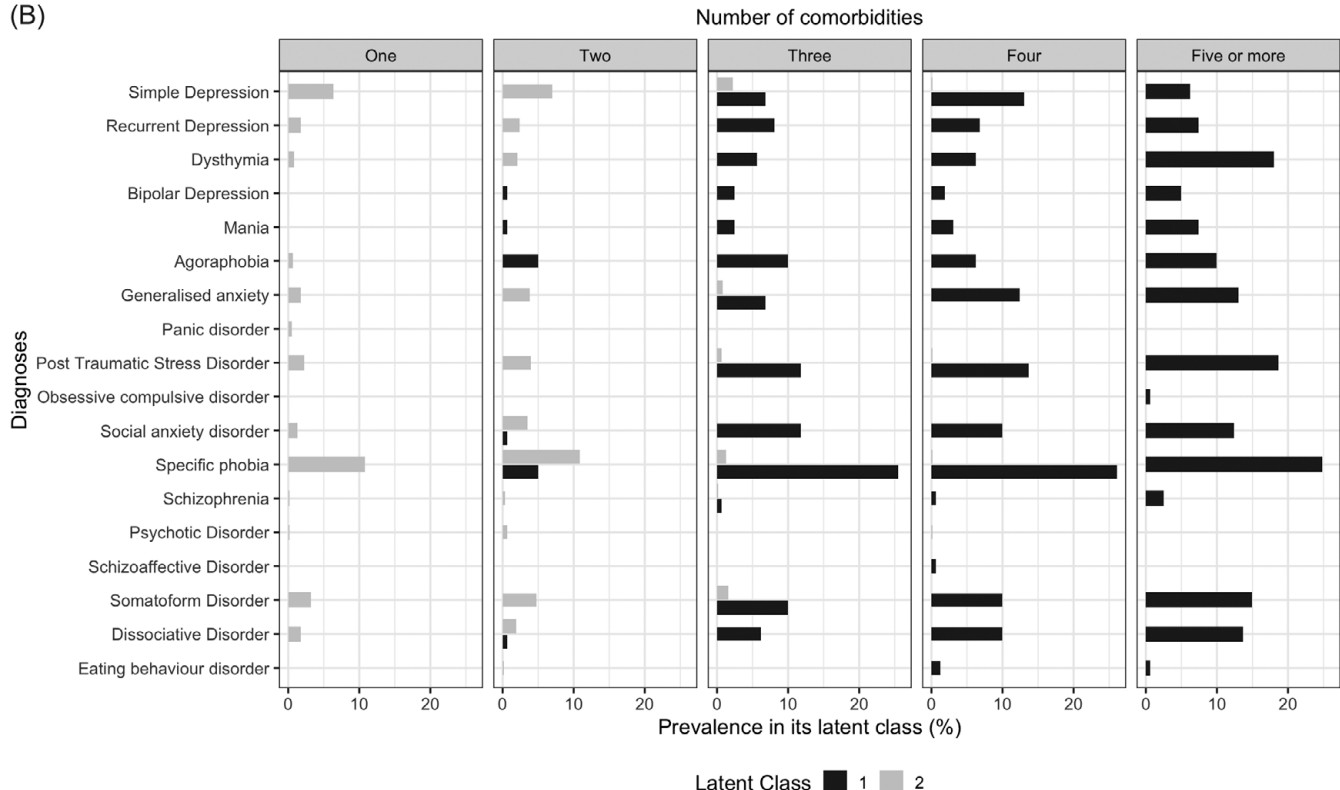

**Figure 1.** (A) Profiles of latent classes of mental health diagnoses. (B) Prevalence in its latent class of mental health diagnoses based on number of comorbidities.

## Discussion

Our study aimed to evaluate the association of the interaction between 5HTTLPR and BDNF genetic polymorphisms with a profile of different mental disorders, using latent class analysis in a Chilean primary care population. Using this methodology, two mental health diagnostic profiles were identified, finding a group with both higher comorbidity and an increased likelihood of certain types of diagnoses. Additionally, associations were found between

**Table 3.** Logistic regression of belonging to the higher psychiatric comorbidity group

| | | Univariate | Interaction | + Biological | + Psychosocial |
|---|---|---|---|---|---|
| BDNF polymorphisms | GG | – | – | – | – |
| | GA-AA | 1.03 (0.73–1.45, p = 0.887) | 0.98 (0.67–1.41, p = 0.896) | 0.99 (0.68–1.44, p = 0.959) | 0.92 (0.61–1.36, p = 0.662) |
| 5HTTLPR polymorphisms | L'/S'-S'/S' | – | – | – | – |
| | L'/L' | **0.54 (0.31–0.91, p = 0.028)** | **0.39 (0.15–0.88, p = 0.038)** | **0.37 (0.14–0.85, p = 0.030)** | **0.33 (0.12–0.77, p = 0.017)** |
| BDNF(GA-AA):5HTTLPR(L'/L') Interaction | Interaction | | 1.75 (0.58–5.69, p = 0.329) | 1.95 (0.64–6.42, p = 0.249) | 2.20 (0.69–7.53, p = 0.192) |
| Sex | Female | | | – | – |
| | Male | | | **0.23 (0.12–0.41, p<0.001)** | **0.23 (0.12–0.42, p<0.001)** |
| Age | Mean (SD) | | | 1.00 (0.99–1.01, p = 0.977) | 1.00 (0.99–1.02, p = 0.507) |
| Educational level | High school | | | | – |
| | Basic education or less | | | | 0.94 (0.59–1.49, p = 0.795) |
| | Higher education | | | | 1.27 (0.66–2.39, p = 0.459) |
| Presence of a family history of depression | No | | | | – |
| | Yes | | | | **2.48 (1.65–3.72, p<0.001)** |
| Number of different forms of abuse experienced in childhood[a] | Mean (SD) | | | | **1.73 (1.44–2.07, p<0.001)** |

[a]Physical, psychological and/or sexual abuse (scale (sum) from 0 to 3).

**Table 4.** Logistic regression of belonging to the higher psychiatric comorbidity group based on 5HTTLPR functional genotype according to BDNF functional genotype

| | | GG | GA – AA |
|---|---|---|---|
| 5HTTLPR polymorphisms | L'/S'-S'/S' | – | – |
| | L'/L' | **0.31 (0.11–0.74, p = 0.014)** | 0.73 (0.33–1.51, p = 0.416) |
| Sex | Female | – | – |
| | Male | **0.18 (0.05–0.45, p = 0.001)** | **0.27 (0.11–0.58, p = 0.002)** |
| Age | Mean (SD) | 1.01 (0.99–1.03, p = 0.420) | 1.00 (0.98–1.02, p = 0.836) |
| Educational level | High school | – | – |
| | Basic education or less | 1.07 (0.56–2.02, p = 0.839) | 0.84 (0.43–1.63, p = 0.598) |
| | Higher education | 1.83 (0.67–4.72, p = 0.223) | 0.94 (0.38–2.16, p = 0.885) |
| Presence of a family history of depression | No | – | – |
| | Yes | **2.08 (1.15–3.71, p = 0.014)** | **2.93 (1.64–5.22, p<0.001)** |
| Number of different forms of abuse experienced in childhood[a] | Mean (SD) | **1.79 (1.37–2.36, p<0.001)** | **1.66 (1.30–2.14, p<0.001)** |

[a]Physical, psychological and/or sexual abuse (scale (sum) from 0 to 3).

belonging to this group and 5HTTLPR polymorphisms, which showed variations in relation to the interaction of these polymorphisms with BDNF polymorphisms.

With respect to health diagnosis profiles, our results revealed a low psychiatric comorbidity group, with 75% of their sample having one or no mental health disorder. In contrast, the other class demonstrated a significantly higher prevalence of comorbidities, with 75% of the individuals presenting with three or more mental health disorders. Importantly, these two groups were drawn from the same sample of primary care patients, but had marked differences in their psychiatric comorbidity profiles. After analysing the diagnostic profiles between these groups, it was found that the higher psychiatric comorbidity class had a higher prevalence of anxiety and stress-related disorders in combination with other

anxiety diagnoses or other types of diagnoses, such as major depression. In contrast, individuals in the low psychiatric comorbidity group, despite having a similar prevalence of major depression as their counterparts, were less likely to have other comorbidities such as generalised anxiety. These differences may be attributed to the specific diagnoses within each class. For example, Blanco et al. (2014) reported in their study that patients diagnosed with major depression and generalised anxiety had distinct patterns of comorbidities. In their study, the group presenting these diagnoses together (similar to our higher psychiatric comorbidity group) was compared with the anxiety disorder alone and major depression alone groups, finding that the latter group had fewer other psychiatric comorbidities, which is consistent with the characteristics of our low psychiatric comorbidity group. The high prevalence of anxiety disorders observed in the higher comorbidity class may also reflect clinical implications within the same group. On the one hand, this could reflect the common existence of risk factors among different anxiety disorders (Stein et al., 2017), in addition to the increased likelihood of comorbidities between anxiety disorders found in different studies (Greene and Eaton, 2016; Stein et al., 2017). For example, the presence of specific phobias has been associated with the development of agoraphobia, and the presence of agoraphobia with the development of generalised anxiety disorder. On the other hand, this may also be attributed to the natural progression of certain disorders, such as the clinical progression of isolated panic attacks into panic disorder, which subsequently develops into agoraphobia (Stein et al., 2017).

The characteristics that emerge from the identification of different clinical profiles of the same general population of primary care patients might otherwise have been invisible without the use of a methodology such as latent class analysis. Furthermore, current diagnostic references based on the fulfilment of clinical criteria defined by experts are challenged as they may not be sufficient to identify the common basis of the different diagnoses that coexist in this type of psychiatric comorbidity profile (20). The development of new forms of taxonomy related to mental disorders, such as the Research Domain Criteria (RDoC) (Insel et al., 2010) and the Hierarchical Taxonomy of Psychopathology (HiTOP) (Kotov et al., 2017), may provide better insight into the common basis of these diagnostic profiles. However, further research using such methodologies and taxonomic systems is needed to corroborate the above statement.

Regarding the correlation between these latent classes and genetic polymorphisms, it was observed that the L'/L' polymorphism of the serotonin transporter, which represents the proper functioning of this transporter (Kourmouli et al., 2013; Mak et al., 2015; Fratelli et al., 2020; Miozzo et al., 2020), was associated with a diminished likelihood of belonging to the group with the highest psychiatric comorbidity when compared to S' carriers. However, when considering the interaction with BDNF, the reduced risk of the L'/L' polymorphism appeared only in the presence of the BDNF GG polymorphism, which would also represent the proper functioning of this neurotransmitter (Terracciano et al., 2010; Gatt et al., 2015; Notaras et al., 2015; Gören, 2016). Biological evidence suggests that there is a synergistic influence between BDNF and 5HTTLPR in the emergence of phenotypes that are more vulnerable to environmental stressors. This is due to altered transcriptional modulation and early neuronal plasticity, which are mutually enhanced by alterations in each of these neurotransmitters (Homberg et al., 2014). Indeed, Aguilera et al. found that childhood sexual abuse experiences had a greater impact on the onset of depressive symptoms in carriers of the valine-to-methionine (GA) substitution allele for

BDNF and the short allele (S'/S') for 5HTTLPR (Aguilera et al., 2009). This vulnerability to stressors also coincides with the fact that the group with the highest psychiatric comorbidity had a higher prevalence of specific phobias and PTSD, in conjunction with other diagnoses. In view of the above, it can be theorised that an alteration in BDNF or 5HTTLPR, due to their synergistic influence, would generate vulnerability profiles to stressors that would be less likely to appear if both neurotransmitters were functioning properly. Although it was not possible to identify a significant RERI when evaluating the interaction on an additive scale, the significant association of the L'/L' polymorphism of the serotonin transporter when compared to S' carriers, identified in the univariate analysis but only discernible in the presence of the BDNF GG polymorphism, may offer insight into the inconsistencies in findings across other studies that have individually associated these polymorphisms with mental health disorders, without considering the functional status of other genes. This emphasises the importance of examining interactions between polymorphisms when investigating genetic and mental health associations, as this could mask the effect of a genetic change and reduce the capacity to detect associations (Cordell, 2002). In fact, the L'/L' polymorphism also showed a tendency to reduce the risk of belonging to the group with the highest psychiatric comorbidity in carriers of the valine-to-methionine substitution in BDNF, but failed to identify a significant association. This could also be due to the additional risk that this BDNF genetic polymorphism would present in the L'/L' group, as can be better seen in Supplementary Appendix 2. Furthermore, this raises the question of whether, in future studies, it would be beneficial to establish a reference group comprising individuals who do not exhibit genetic risk alterations.

Statistically significant results were also found for certain biological and psychosocial variables studied. The study of the gene–pathology sequence also requires consideration of environmental influences, as mentioned by du Pont et al. (2019). The increased risk of belonging to this group in a population with a family history of depression could be related to a possible genetic inheritance and/or an environmental factor to be considered (Monroe et al., 2014; Kendall et al., 2021), the latter also making sense in relation to the increased risk if more forms of childhood maltreatment are identified (Clarke et al., 2020; Goemans et al., 2023). It is beyond the scope of this article to examine how these factors may influence or ultimately trigger the risk associated with genetic polymorphism profiles. On the other hand, the lower risk of belonging to the profile of greater psychopathology in the male sex is consistent with previous research where the presence of these disorders is higher in the female population (Maestre-Miquel et al., 2021).

Essau and De la Torre-Luque promote the use of LCA for the study of complex multidimensional phenomena, mentioning that characterising the profile of psychiatric comorbidities allows a higher-order classification that could have important clinical implications (Essau and de la Torre-Luque, 2019). The identification of clinical profiles based on combinations of genetic polymorphisms could optimise early screening processes from primary health care, given its role as a gateway for the detection and early treatment of mental disorders (Moscovici et al., 2020), thus addressing the barriers to appropriate treatment perceived by certain groups of patients (Jorm et al., 2017), particularly those with more complex clinical profiles who may not receive the most appropriate treatment in primary care (Moscovici et al., 2018). This could facilitate timely referral or, on the other hand, the implementation of "intervention packages" from primary health care, such as the one developed in Mathew's article (Mathew, 2022),

where they demonstrated the benefit of implementing a structured coordination of psychological interventions in patients with anxiety disorders (a profile similar in clinical terms to that of the group with the highest psychiatric comorbidity found in our article).

Among the limitations of this study, it should be mentioned that the prevalences obtained may not represent the current situation, as the CIDI scale was applied in 2003; however, the aim of this study was not to determine the specific prevalence of each diagnosis in the Chilean population, but to define psychiatric comorbidity profiles, which were established with lifetime prevalences. Additionally, due to the structure of the original database, it is only possible to determine the presence or absence of the respective diagnoses, without clear information on the severity that some of them might have. Conversely, there is a possibility that solutions with a larger number of latent classes are relevant for a more extensive exploration of psychiatric comorbidity profiles. The decision to maintain two latent classes in our study, despite the identification of well-defined and reasonable clinical profiles, was also influenced by the need to maintain adequate sample sizes for the remaining analyses. The evaluation and characterisation of other profiles with a larger number of latent classes would be a fruitful avenue for future research, though it is beyond the scope of this article. One of the strengths of this study is that the cohort from which this study is derived corresponds to one of the largest Chilean samples of diagnostic prevalence, within which we were able to maintain a significant subgroup in terms of quantity, from which saliva samples could be taken for genotyping. On the other hand, to the best of our knowledge, our study is one of the few Latin American studies to establish relationships between functional genotypes and diagnostic comorbidity profiles using latent class methods and taking into account interactions between polymorphisms.

## Conclusion

In conclusion, this study identified an association between the 5HTTLPR and BDNF polymorphisms, particularly in the combination where both polymorphisms exhibited normal functioning with a reduced risk of belonging to a profile of increased psychiatric comorbidity. These findings underscore the need to consider the functionality of other polymorphisms when evaluating their association with psychopathology. On the other hand, and in relation to psychiatric comorbidity profiles, the use of latent classes was shown to be a promising tool for identifying, within a common sample, subgroups that may have significant clinical implications, highlighting the necessity for further research to transcend the confines of individual diagnoses. This should encompass the relationships and natural history of distinct mental disorders, which may be pivotal in investigating the interconnections between genetics and mental health. Furthermore, this line of research has the potential to yield advancements in the taxonomy of mental health diagnoses.

**Open peer review.** To view the open peer review materials for this article, please visit http://doi.org/10.1017/gmh.2025.10062.

**Supplementary material.** The supplementary material for this article can be found at http://doi.org/10.1017/gmh.2025.10062.

**Data availability statement.** Data will be uploaded to the institutional Dataverse repository on publication. https://doi.org/10.48665/udec/HIE40I.

**Acknowledgements.** We thank Mr. Silverio Torres for the initial preparation of data for further analyses.

**Author contribution.** All authors made substantive contributions to the study. The individual contributions to the paper are as follows. Conceptualization: E. Moraga-Escobar, K. Schmidt, B. Vicente, R. Rojas-Ponce. Data curation: B. Vicente, R. Rojas-Ponce, J. Castillo-Navarrete, E. Moraga-Escobar, K. Schmidt. Formal analysis: E. Moraga-Escobar, K. Schmidt. Funding acquisition: B. Vicente, R. Rojas-Ponce. Investigation: B. Vicente, R. Rojas-Ponce, J. Castillo-Navarrete. Methodology: E. Moraga-Escobar, K. Schmidt, B. Vicente, R. Rojas-Ponce. Writing—original draft: E. Moraga-Escobar, K. Schmidt. Writing—review & editing: E. Moraga-Escobar, K. Schmidt, B. Vicente, R. Rojas-Ponce, J. Castillo-Navarrete, X. Macaya, A. Guzman-Castillo, P. Lagos.

**Financial support.** Agencia Nacional de Investigación y Desarrollo (ANID): Fondo de Fomento al Desarrollo Científico y Tecnológico #DO2I-1140 (BV); Agencia Nacional de Investigación y Desarrollo (ANID): Fondo Nacional de Desarrollo Científico y Tecnológico #1110687 (BV); Universidad de Concepción VRID: #218.087.043-1.0 (RR; BV).

**Competing interests.** All authors confirm that they have no conflicts of interest to report.

**Ethics statement.** All components of the study were approved by the institutional Research Ethics Committees; the associated confirmation letters can be provided upon request.

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
