## [Reviewer Report]

This is an interesting study about an important topic, with an innovative but sound methodology.

A few minor recommendations for the authors:

1. The interaction between 5HTTLPR and BDNF polymorphisms have been reported previously in another sample of Chilean patients, in a study of neuroticism trait in borderline personality disorders (Salinas et al., SERT and BDNF polymorphisms interplay on neuroticism in borderline personality disorder, BMC Res Notes. 2020 Feb 7;13(1):61. doi: 10.1186/s13104-020-4924-6.). This could be considered whether in the introduction or in the discussion sections.

2. In methods, please clarify what kind of sample (if any) was “drawn” from adults in 2003 (or consider rephrasing this sentence for a better understanding). Also, please clarify if samples provided between 2011 and 2012 are part of the same FONDEF study, or if these were part of a different study (if so, please name which one).

3. Please explain at the begining of the results why the final sample consisted of 789 participants, given the larger number of participants from the studies reported in methods.

4. Table 4 needs to be renamed from “Tabla 4” to “Table 4”.

---

## [Reviewer Report]

The authors demonstrated that L‘/L’ polymorphism of the 5HTTLPR gene was associated with a reduced risk of belonging to the high-comorbidity group, particularly when paired with the GG polymorphism of the BDNF gene. The research and results are interesting, but reviewer found some issues that have to be answered before publication can be recommended.

1. The Information about primary care patients is not clear. It is necessary to clarify the reasons of the patients who visited this primary care centre, such as their diseases, health check or vaccinations and so on.

2. The author wrote that the initial study was conducted in 2003 and subsequent study was done between 2011 and 2012. What does the difference in the years in which the studies were conducted mean ?.

3. It would be worthwhile examine the correlation between the number of comorbidities and severity of the psychiatric disease. It would be desirable if any suggestive results on these issues were presented.

4. In genetic research, racial differences are often recognized as a limitation of the study. Since this study was conducted on Chileans, it would be better to add information about the characteristics of the participants.

---

## [Editor Report]

Dear authors:

Your manuscript needs a Major Revision. Please, follow the suggestions of reviewers.

Additionally, you must clarify other issues of the manuscript:

The interaction of two or more genes is called epistasis; then you need to address this concept in the study.

In the methods, it is not clear why you are using two latent classes. The literature on psychiatric comorbidity is extensive; suggesting 3 or 4 latent classes, thus this kind of analyses must be theory-concept driven instead of data driven. 

For example, see: Burmeister M McInnis M G Zöllner S 2008 Psychiatric genetics progress amid controversy Nature Reviews Genetics 9,7; 527-540; and Caspi, A Moffitt, T E 2018 All for one and one for all Mental disorders in one dimension American Journal of Psychiatry, 175,9; 831-844. 

In relation to the serotonin transporter SLC6A4 promoter, there is a 5-HTTLPR-rs25531 mini-haplotype which implies the functional division of individuals into three expression types, each corresponding to one, two or four diplotypes. Did you have data on the diplotypes? Please, discuss it.

In relation to the logistic regression analyses, the methods did not explain well the “other” variables in such analyses. Why did you included this additional set of variables? Also, the interaction terms are in the multiplicative scale which has very little power for detecting gene X gene or gene X environment interactions, given your sample size. 

These interactions must be addressed with the additive scale. Please, see: VanderWeele, T. J., & Knol, M. J. (2014). A tutorial on interaction. Epidemiologic methods, 3(1), 33-72. and VanderWeele, T. J. (2015). Explanation in causal inference: methods for mediation and interaction. Oxford University Press.

Additionally, in the logistic regression you are using “childhood abuse” as a main effect variable; but, the literature is clear on the gene X environmental interaction between childhood abuse and the 5-HTTLPR polymorphism. You must address this issue in the analyses and discussion.

Best regards.

The Handling Editor.

---

## [Reviewer Report]

The authors have responded to the previous comments which have been adequately corrected and publication can be recommended.

---

## [Reviewer Report]

The authors have carefully included all recommendations, and the manuscript is now almost ready to be published. They have clarified the study protocol in a way that is now easy to understand for readers that are not familiar with the study process.

One minor detail to be addressed before publication: in lines 30 and 31 please change “in a study of Chilean university students” for “in a previous Chilean study that included patients with borderline personality disorder” (since it was not based on university students).

---

## [Editor Report]

Dear Authors:

In addition to the minor revisions requested by the reviewers, please, add some explanations in relation to the RERI additive interaction index, in the “methods section”. The scientific literature has some examples of additive interaction studies in relation to the 5HTTLPR polymorphisms.

Best regards.

---

## [Editor Report]

Dear authors: 

The reviewers accepted your corrections. 

Now, the manuscript is accepted. 

Best regards.